# Short Peripheral Venous Catheters Contamination and the Dangers of Bloodstream Infection in Portugal: An Analytic Study

**DOI:** 10.3390/microorganisms11030709

**Published:** 2023-03-09

**Authors:** Nádia Osório, Vânia Oliveira, Maria Inês Costa, Paulo Santos-Costa, Beatriz Serambeque, Fernando Gama, David Adriano, João Graveto, Pedro Parreira, Anabela Salgueiro-Oliveira

**Affiliations:** 1Polytechnic Institute of Coimbra, Coimbra Health School, 3046-854 Coimbra, Portugal; 2Molecular Physical-Chemistry, University of Coimbra, 3004-535 Coimbra, Portugal; 3Health Sciences Research Unit: Nursing (UICISA: E), Nursing School of Coimbra, 3004-011 Coimbra, Portugal; paulocosta@esenfc.pt (P.S.-C.);; 4Centro Hospitalar e Universitário de Coimbra, 3004-561 Coimbra, Portugal

**Keywords:** peripheral intravenous catheter, *Staphylococcus* spp., virulence factors, antibiotic resistance

## Abstract

Peripheral venous catheters (PVCs) are the most used vascular access devices in the world. However, failure rates remain considerably high, with complications such as PVC-related infections posing significant threats to patients’ well-being. In Portugal, studies evaluating the contamination of these vascular medical devices and characterizing the associated microorganisms are scarce and lack insight into potential virulence factors. To address this gap, we analyzed 110 PVC tips collected in a large tertiary hospital in Portugal. Experiments followed Maki et al.’s semi-quantitative method for microbiological diagnosis. *Staphylococcus* spp. were subsequently studied for the antimicrobial susceptibility profile by disc diffusion method and based on the cefoxitin phenotype, were further classified into strains resistant to methicillin. Screening for the *mec*A gene was also done by a polymerase chain reaction and minimum inhibitory concentration (MIC)-vancomycin as determined by E-test, proteolytic and hemolytic activity on skimmed milk 1% plate and blood agar, respectively. The biofilm formation was evaluated on microplate reading through iodonitrotetrazolium chloride 95% (INT). Overall, 30% of PVCs were contaminated, and the most prevalent genus was *Staphylococcus* spp., 48.8%. This genus presented resistance to penicillin (91%), erythromycin (82%), ciprofloxacin (64%), and cefoxitin (59%). Thus, 59% of strains were considered resistant to methicillin; however, we detected the *mec*A gene in 82% of the isolates tested. Regarding the virulence factors, 36.4% presented α-hemolysis and 22.7% β-hemolysis, 63.6% presented a positive result for the production of proteases, and 63.6% presented a biofilm formation capacity. Nearly 36.4% were simultaneously resistant to methicillin and showed expression of proteases and/or hemolysins, biofilm formation, and the MIC to vancomycin were greater than 2 µg/mL. Conclusion: PVCs were mainly contaminated with *Staphylococcus* spp., with high pathogenicity and resistance to antibiotics. The production of virulence factors strengthens the attachment and the permanence to the catheter’s lumen. Quality improvement initiatives are needed to mitigate such results and enhance the quality and safety of the care provided in this field.

## 1. Introduction

In healthcare, the majority of the patients admitted to a hospital require a vascular access device to comply with the intended therapeutic plan [1,2,3,4]. There are several types of vascular access devices in use, such as short and long peripheral venous catheters (PVC), midline catheters, peripherally inserted central catheters (PICCS), and central venous catheters (CVCs). Each year, it is estimated that more than two billion PVCs are used worldwide, constituting the most common type of vascular access device in clinical settings [3,4,5,6,7,8,9]. In Portugal, the latest national evidence suggests that 51.9% (Madeira islands) to 72.7% (Azores Islands) of all hospitalized patients will require at least one PVC to complete their therapeutic plan [10]. This data suggests that PVCs are the most commonly found invasive device in Portuguese hospitals, above urinary catheters (17.6–28.5%) and CVCs (5–10%) [10].

A PVC is a small, flexible tube inserted in a peripheral vein, mainly the metacarpal, cephalic, or basilica vein, and secured to the skin with an adhesive dressing [4,5,11]. These devices are typically made of polyurethane, and their size can range from 26 to 14 Gauge (G) [4,5]. This medical device is ideally suited for short-term use (e.g., 72–96 h), mainly indicated in the delivery of intravenous fluids and drugs within osmolarity and pH levels considered appropriate for peripheral veins [4,11,12].

Although rather simplistic in design and technique, peripheral intravenous catheterization is an invasive procedure that can lead to local (e.g., phlebitis, infiltration, hematoma, nerve puncture) and systemic complications (e.g., catheter-associated bloodstream infection [CRBSI], tip fracture and migration) [3,4,6,8,9,13,14,15,16,17,18]. These complications lead to premature catheter failure and an increase in morbimortality rates, workload for healthcare professionals, admission periods, and total care costs for the healthcare system [2,4,6,14,15,18].

The insertion of vascular access devices is a potential pathway for the entry of microorganisms into the bloodstream, which can lead to CRBSIs [11,13,19]. However, some recent studies showed that the PVC-related bloodstream infections (PVCR-BSIs) rates (0.1%, 0.5 per 1000 catheter-days) are lower than the other intravascular devices, such as CVCs (4.4%, 2.7 per 1000 catheter-days) and PICCs (2.4%, 2.1 per 1000 catheter-days) [7,20,21]. Despite this, the rates of PVCR-BSIs may rise in the future due to the wide use of PVCs [1,21,22]. These are still responsible for 5% (670 per 100.000 patients) of nosocomial bacteremia, being implicated in the etiology of hospital-acquired infections (HAIs) [21,22,23].

For the occurrence of CRBSIs, three pathways are described for the entry of microorganisms through the medical device from a non-sterile external environment into the normally sterile bloodstream [23,24,25]. The first is called extraluminal, where the migration of microorganisms occurs mainly from the patient’s skin into the catheter tract. This process may occur during the insertion of the catheter or while the catheter is in situ. However, it is the most common route of infection for short-term catheters. The second route is called intraluminal, involving direct contamination of catheter hubs and connectors by contact with the hands of health professionals who handle it, contaminated fluids or devices. The third contamination route is when the catheter is contaminated by microorganisms circulating in the bloodstream when there is already a preexisting infectious condition responsible for the contamination of the device [22,23].

These medical devices provide a surface area to which microorganisms can attach [23]. The most common microorganisms involved are *Staphylococcus* species, namely coagulase-negative staphylococci (CoNS), predominantly *S. epidermidis* and *S. aureus* [21,22,23,26,27,28,29]. Although these are commensal human skin bacteria and are considered non-pathogenic, they have been recognized as relevant opportunistic pathogens [22,27,29]. Other microorganisms have been identified as the Gram-negative bacilli, *Enterococcus* spp., and fungi, such as *Candida* species [21,22,26,27]. Some of these microorganisms are associated with hospital-acquired infections through cross-contamination between health professionals/medical devices and the patients, mainly due to the incorrect disinfection technique of the catheter insertion site, poor hand hygiene practices, and inefficient device maintenance [19,24,27,30].

The *Staphylococcus* genus is the most common cause of indwelling device-associated infections and nosocomial and community-acquired infections. Given the increasing use of PVCs and the global threat represented by the increase in antibiotic resistance and the virulence of these microorganisms, the treatment of these infections becomes difficult, prolonged, and ineffective [31,32,33].

Despite its ubiquity across Portuguese healthcare settings, contrary to other international settings, there is little known evidence of PVC contamination, with the country lacking representation in well-known multicentric studies conducted recently in this field [34,35,36]. Therefore, this study aimed to evaluate the microbial contamination of PVCs used in a large tertiary hospital in Portugal, identifying the most prevalent microorganisms and evaluating the risk associated with these contaminations as seen by evaluating their virulence factors and antibiotic resistance.

## 2. Materials and Methods

### 2.1. Sample Characterization

The present study was performed in a cardiology ward of a large tertiary hospital in the central region of Portugal. It had the approval of the Ethics Hospital Committee (reference number 0226/CES) and authorization number 14037/2017 from the Portuguese Data Protection Authority. Written informed consent was obtained during study recruitment. A total of 110 PVC tips (2 cm) were collected and were in sterile flasks at 4 °C. Samples were then transported for laboratory analysis at the Coimbra Health School—Polytechnic Institute of Coimbra.

### 2.2. Microbiological Analysis

#### 2.2.1. Isolation and Identification

The PVCs tip samples were inoculated on a Columbia agar base (supplemented with 5% of sheep blood) using the technique of Maki et al. [37]. The cultures were incubated at 37 °C in the normal atmosphere for 18–24 h. After enumeration and macroscopic evaluation, we performed the isolations of the macroscopically different microorganisms in Tryptic Soy Agar.

Pure colonies obtained were characterized by the Gram staining and biochemical tests as catalase and/or oxidase to a primary identification. Subsequently, we used biochemical identification galleries as API Staph (REF 20500) bioMérieux^®^, API 20 Strepto (REF 20600) bioMérieux^®^, API 20 NE (REF 20050) bioMérieux^®^, API 20 E (REF 20100) bioMérieux^®^, following the manufacturer’s instructions. The identification of the species was obtained using the Apiweb software as well as the score of identification.

#### 2.2.2. Detection of Extracellular Enzymes

The extracellular protease production was determined by the plate assay in Luria Bertani (Merck, Darmstadt, Germany) agar medium supplemented with 1% of skimmed milk (*w*/*v*). A bacterial suspension of 0.5 McFarland was prepared and subsequently inoculated (5 μL), then all plates were incubated at 37 °C for 24 h in a normal atmosphere. The presence of extracellular proteases was revealed by the formation of clear halos around the colonies which were measured. The halos were classified as negative (−) in the absence of a halo, as weak positive (+/−) in the presence of a halo less than 11 mm, as positive (+) in the presence of a halo less than 13 mm, and as strongly positive (++) in the presence of a halo less than 15 mm [38].

The hemolytic activity was determined by the plate assay using a Columbia Agar with 5% sheep blood (Merck, Darmstadt, Germany). A bacterial suspension of 0.5 McFarland was prepared, and subsequently, 5 μL was inoculated and plates incubated at 37 °C for 24 h in a normal atmosphere. The production of hemolysins was identified by the presence of clear (β-hemolysis) or diffuse (α-hemolysis) halos around the colonies. The absence of a halo shows that there was no production of hemolysins [39].

#### 2.2.3. Biofilm Formation Assay

From a 0.5 McFarland bacterial suspension, 10 μL was inoculated into a 96 wells plate with 100 μL of Luria Bertani Broth (5 replicates for each strain were done), and the incubation was performed at 37 °C overnight with a normal atmosphere. The planktonic cells were transferred, and 25 μL of 0.2 g·L^−1^ iodonitrotetrazolium chloride 95% (INT) solution was added. Cells from biofilms were washed from multiwells with 200 μL of phosphate-buffered saline (PBS 1×) to remove all non-adherent cells, and this process was repeated 2 more times. A 100 μL of fresh media was added to the correspondent wells after rinsing, followed by the addition of 25 μL of 0.2 g·L^−1^ INT solution. Both plates were immediately covered with aluminum foil and incubated in the dark at 37 °C. The reading was performed at λ: 492 nm after 30 min of incubation using the microplate reader (Thermo Fisher Scientific, USA). The biofilm formation results were normalized using the ratio of adherent cells at OD492nm/planktonic cells at OD492nm. The isolates which have values below 0.75 were classified as moderate biofilm-formers, values between 0.75 and 1.0 were classified as high biofilm-formers, and values above 1.0 were classified as very high biofilm-formers [40].

#### 2.2.4. Antimicrobial Susceptibility Test by the Disk Diffusion Method

The evaluation of the antimicrobial susceptibility profile of the isolates obtained was performed by the disk diffusion method (modified Kirby-Bauer’s test). A suspension of 0.5 McFarland in sterile NaCl 0.9% from the fresh and pure culture was obtained and then inoculated in Muller Hinton agar. The selection of antimicrobial disks for *Staphylococcus* spp. considered its clinical application: cefoxitin (30 μg), ciprofloxacin (5 μg), chloramphenicol (30 μg), erythromycin (15 μg), gentamicin (10 μg), penicillin (10 units), tetracycline (30 μg), and trimethoprim-sulfamethoxazole (1.25/23.75 μg). The antibiogram was incubated in a normal atmosphere at 37 °C for 18–24 h. The reading was based on the inhibition halos (mm) measurement, and phenotypically the strain was classified as sensitive, intermediate, and resistant. The interpretation of the results was performed, taking into account the Clinical and Laboratory Standards Institute (CLSI, 2018) [41].

The detection of methicillin resistance was also performed, based on the reading of the inhibition halo to cefoxitin, taking into account CLSI standards [41].

#### 2.2.5. Determination of Susceptibility to Vancomycin by E-Test Method

The susceptibility to vancomycin was performed by the E-test method with the determination of the minimum inhibitory concentration (MIC). A suspension of 0.5 McFarland in sterile NaCl 0.9% from the fresh and pure culture was obtained and then inoculated in Muller Hinton agar. The antibiogram was incubated in a normal atmosphere at 37 °C for 18–24 h. The results were interpreted according to CLSI, 2018 [41].

#### 2.2.6. DNA Extraction

A bacterial cell suspension, with one pure colony, in LB medium was incubated overnight at 37 °C. After the inoculum was centrifuged for 10 min at 13,000× *g*, then the supernatant was discarded, and the pellet was resuspended in 200 µL of Tris-EDTA. The DNA extraction was made using the GeneJET Genomic DNAPurification Kit #K0721 (Thermo Fisher Scientific, Waltham, MA, USA) following the manufacturer’s instructions. The DNA solution obtained was stored at −20 °C.

#### 2.2.7. Detection of *mec*A Gene Using PCR Technique

The *mec*A gene amplification was performed by PCR using specific primers: *mec*A-R (5′-CAATTCCACATTGTTTCGGTC-3′) and *mec*A-F (5′-GAAATGACTGAACGTCCGATA-3′) (Metabion International AG, Planegg, Germany). The primers used were designed using the nucleotide sequence databases from NCBI. Amplification was done using 12.5 μL DreamTaq PCR Master Mix (2X) (Thermo Scientific, Waltham, MA, USA, EUA), 10 pmol of the forward and the reverse primers, 9.5 μL of the nuclease-free water, and 1 μL of the bacterial DNA. The PCR reactions were performed using a MyCycler Thermal Cycler (Bio-Rad, Hercules, CA, USA). The amplification conditions consisted of an initial denaturation step (95 °C for 5 min), followed by 30 amplification cycles consisting of denaturation (94 °C for 45 seg), annealing (53 °C for 45 seg), and extension (72 °C for 1 min) and a final extension step (72 °C for 10 min). The DNA of positive and negative strains of the *mec*A gene was used as positive and negative controls previously characterized and provided by a hospital. The reaction products were separated by electrophoresis on a 1.5% (*w*/*v*) agarose gel. All gels were run in 1×TAE buffer at 80 V for 80 min, stained in 0.5 μgmL^−1^ of ethidium bromide solution, and the images were acquired with the Gel Doc XR+ System (Bio Rad, Hercules, CA, USA).

## 3. Results

### 3.1. Prevalence of the PVCs Microbiological Contamination and Identification of the Isolates

Patients had an average age of 79 (±11) years. Concerning the 110 PVC tips, 30% were contaminated, from which 45 macroscopically different isolates were obtained. The most prevalent genus was *Staphylococcus* spp., with 48.8%. Belonging to this genus, isolated bacterial species were *S. aureus* (4.4%), S. *epidermidis* (26.7%), *S. haemolyticus* (11.1%), *S. lentus* (2.2%), *S. warneri* (2.2%), and *S. xylosus* (2.2%). Other clinically relevant microorganisms were found as *Enterococcus* spp. (4.4.%), *Escherichia coli* (2.2%), *Klebsiella pneumoniae* (2.2%), and *Stenotrophomonas maltophilia* (2.2%). The remaining 40% corresponded to other microorganisms whose identification success rate was low. In 8.2% of the infected PVCs, more than one isolate was observed (Table 1).

### 3.2. Virulence Factors in Staphylococcus Isolates—Proteolytic and Hemolytic Activity and Biofilm Formation

Concerning protease production, most of the isolates displayed a positive phenotype. However, 40.9% of the isolates presented higher levels of proteolytic activity. In the hemolytic activity, different phenotypes were observed; some strains were negative (40.9%), 36.4% presented α-hemolysis, and 22.7% presented β-hemolysis. All strains demonstrated the ability to produce biofilm, with ratios ranging from capacity 0.45–1.7. However, it was found that the majority (63.8%) had high to very high capacity. It was also observed that of this majority, 50% of the isolates correspond to *S. epidermidis* (Table 2).

### 3.3. Antibiotic Resistance in Staphylococcus Isolates: Susceptibility Profile and Presence of mecA Gene

*Staphylococcus* isolates were mainly resistant to penicillin (91%), erythromycin (82%), ciprofloxacin (64%), and cefoxitin (59%). The antimicrobial agents for which they presented greater sensitivity were ciprofloxacin (95%), tetracycline (86%), gentamicin (73%), and trimethoprim-sulfamethoxazole (59%) (Figure 1). According to CLSI (2018), we found 59% of methicillin resistance in *Staphylococcus* isolates (65). However, when we evaluated the presence of the genetic determinant *mec*A among methicillin-resistant isolates, putatively encoding PBP2a, we found 82% of positive strains.

Relatively to the vancomycin susceptibility profile, we found sensitivity in all isolates, according to the CLSI classification, and had been into account the identification as *S. aureus* or CoNS (65). However, when we observed the MIC levels of sensitivity, most of the isolates presented values higher than 2 μg/mL.

In addition, it is noted that about 45.5% of methicillin-resistant staphylococci (MRS) and 9.1% of methicillin-susceptible staphylococci (MSS) have MIC for vancomycin greater than 2μg/mL.

## 4. Discussion

Overall, 30% of the analyzed PVCs were contaminated, an alarming result given that in Portugal, this vascular access device is known to be used in an array of clinical settings [42,43,44], with different patient cohorts from neonates to older adults [45,46,47,48], and for a myriad of therapeutic purposes such as intravenous drug and fluid therapy, chemotherapy, contrast administration, and blood sampling [49,50,51,52,53].

Regarding the identified microorganisms, it was verified that the most prevalent genus was *Staphylococcus* in 48.8%, predominantly CoNS (44.4%), of which *S. epidermidis* was the most common (26.7%). Additionally, in these studies, the genus *Staphylococcus* is responsible for 58% of the infections caused by PVCs, of which about 25% are coagulase-negative [21,26]. Other identified microorganisms belonging to this genus were also found in the present investigation: *S. aureus*, *S. haemolyticus*, *S. lentus*, *S. warneri*, and *S. xylosus*.

The occasional presence of *Enterococcus* spp., *E. coli*, *Klebsiella pneumoniae*, and *Stenotrophomonas maltophilia* was also observed. The low incidence can be explained by the fact that they are not part of the normal microbiota of the skin. However, the presence of these microorganisms in a hospital environment can allow their presence in transient microbiota [22]. However, we highlight the relevance of these microorganisms found in the context of healthcare-associated infections (HAIs) as opportunistic pathogens [31]. There are reports of studies on the presence of these species associated with PVC infections, also with low prevalence [21,22,26].

The presence of more than one isolate occurred in 8.2% of contaminated PVCs, this result being similar to that found by Guembe et al., who found only 2.9% of PVCR-BSIs polymicrobial episodes [26].

The higher prevalence of *S. epidermidis* in these medical devices can be due to its commensal feature of the human skin that can be associated with procedures during the insertion and manipulation of PVCs that can lead to infection. In addition, these microorganisms produce virulence factors involved in processes such as attachment to the catheter with increased expression of teichoic acids, adhesins, and autolysins and to the human matrix with increased expression of microbial surface components recognizing adhesive matrix molecule and tissue invasion with the production of extracellular enzymes such as proteases and hemolysins [31,33,54,55]. This fact justifies the results obtained in this study for the proteolytic and hemolytic activity of the isolates, in which protease production was observed in 40.9% and α-hemolysis production in 36.4%.

After adaptation to the catheter, the production of biofilm by these microorganisms is quite common, as verified in 63.8% of the studied isolates that revealed a high capacity of biofilm production. In addition, one species that demonstrated this capacity was *S. epidermidis* in 50% of the isolates. This microorganism is described as the major nosocomial pathogen associated with implanted medical device infections due to its ability to form the extracellular polysaccharide matrix, which allows its protection and strengthens attachment to the catheter [29,32]. The investigation by Hashem et al. shows that 55% of *S. epidermidis* associated with CRBSI are biofilm producers, agreeing with the one observed in our study [56].

Due to the protective nature of the biofilm, associated bacteria are intrinsically resistant to many antibiotics, which can increase up to 1000 times. The main reasons for this may be the difficulty of biofilm penetration by antibiotics, the low growth rate of bacteria, and the presence of antibiotic degradation mechanisms [57,58]. In addition, biofilm promotes horizontal gene transfer between bacteria, causing the spread of drug resistance determinants and other virulence factors [58,59]. This association has already been demonstrated by some studies, such as Belbase et al. and Ghasemian et al., who found higher rates of resistance to multiple drugs and resistance to methicillin in biofilm-producing-strains compared to non-biofilm producing strains [58,60]. The present study corroborated this information, with 59% of isolates being methicillin-resistant *Staphylococcus* and 82% of strains positive for the presence of the *mec*A gene. Mutations in other proteins involved with PBP2a expression, for example, BLIP-II, were able to weakly bind and inhibit PBP2a [61]. The PBP2a has strict substrate requirements; consequently, any factors that influence the formation of the substrate have the potential to perturb or modulate methicillin resistance. Methicillin resistance is affected by the inactivation of genes that affect the autolytic enzyme activities of the cell, such as the llm gene. Activities of the global regulator proteins such as Sar, Agr, and SigmaB are also known to affect methicillin resistance [62]. The presence of this genetic determinant in the majority of the isolates suggests the propensity for the dissemination of resistance genes between bacteria living in biofilm, as evidenced by other studies [29,58,60]. Kitao et al. also indicate that an increase of *S. epidermidis* resistant to methicillin with the presence of the *mec*A gene and capable of biofilm formation has occurred in many cases of CRBSIs [29].

The studied *Staphylococcus* spp. isolates have shown an antimicrobial multi-resistance profile: penicillin (91%), erythromycin (82%), ciprofloxacin (64%), and cefoxitin (59%). This type of profile leads to difficulties in treatment, resulting in prolonged treatment, extended duration of hospital admission, development and persistence of chronic infectious diseases in local and/or distant organs, or even mortality [29].

When methicillin resistance was detected in *Staphylococcus* spp., the glycopeptide antibiotics, such as vancomycin, were selected as the gold standard for the treatment of serious infections caused by these microorganisms. However, a slow but steady increase in vancomycin MIC has been observed in recent years [31]. In this study, all the isolates showed a sensitivity profile to vancomycin. Nevertheless, it is noted that 63.6% presented values higher than 2 μg/mL, which is close to an intermediate phenotype. Thus, this result agrees with the study conducted by Cherifi et al., who also did not observe any resistance to this antibiotic glycopeptide [63]. In clinical settings, vancomycin is used as the treatment of choice and the last resort for infections caused by *Staphylococcus* spp. However, its excessive intravenous use has allowed the adaptation of these microorganisms, causing strains with greater sensitivity to vancomycin [31].

Overall, 36.4% of the strains showed methicillin resistance, MIC-vancomycin greater than 2 μg/mL, the ability to produce at least one extracellular enzyme, and biofilm production. Such results are worrying due to the clinical repercussions that such strains can pose to patients, significantly burdening healthcare staff and institutions.

## 5. Conclusions

We identified microorganisms that colonized PVCs retrieved from a Portuguese hospital ward, mapping their antimicrobial susceptibility profile and their virulence factors. To the best of our knowledge, this is the first study of such nature to be conducted and publicly disseminated in the country.

The main microorganisms were found to belong to the genus *Staphylococcus*. CoNS were the most prevalent, with the presence of *S. epidermidis* in 27.7%. These microorganisms produced virulence factors such as proteases, hemolysins, and biofilm that facilitate their adhesion and permanence in the catheter, often leading to CRBSIs. Associated with these virulence factors, they develop resistance to antimicrobials, making it difficult to choose the trait and reducing the available therapeutic options. By evaluating the production of virulence factors and the susceptibility profile to antimicrobials, we were able to show that the *Staphylococcus* genus demonstrates a great capacity for adaptation and colonization of medical devices, such as PVCs, leading to a greater risk of infection and treatment difficulties.

Our findings substantiate the need to implement quality improvement projects in Portuguese hospitals, focusing on structural and procedural variables that impact the effectiveness and safety of the care provided to patients who require a PVC. Likewise, Portuguese health authorities must implement a specific nationwide surveillance program on PVCR-BSIs, disclosing the annual results found, similar to what is now done for other invasive medical devices (e.g., CVC, urinary catheters).

## Figures and Tables

**Figure 1 microorganisms-11-00709-f001:**
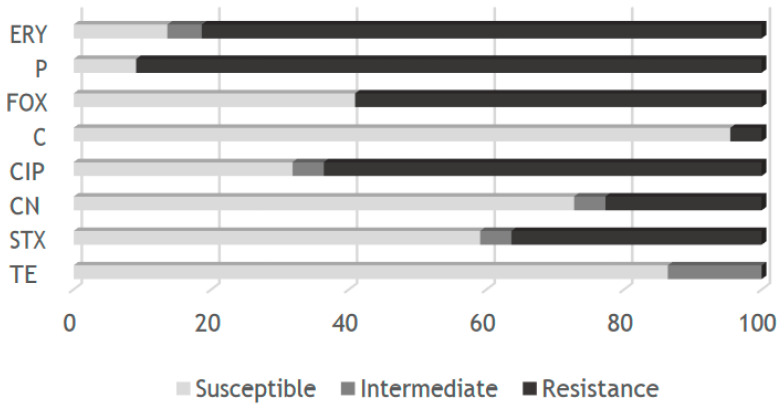
Antibiotic susceptibility profile of *Staphylococcus* isolates; TE-tetracycline; STX—Trimethoprim-Sulfamethoxazole; CN—Gentamicin; CIP—Ciprofloxacin; C—Chloramphenicol; FOX—Cefoxitin; P—Penicillin; ERY—Erythromycin.

**Table 1 microorganisms-11-00709-t001:** Prevalence of isolated microorganisms in the studied PVCs.

Isolated Microorganisms		Isolates	Prevalence (%)
*Staphylococcus* spp.	*S. aureus*	12,17	4.4
	*S. epidermidis*	4–7,11,13–15,18,20–22	26.7
	*S. haemolyticus*	2,8–10,19	11.1
	*S. lentus*	3	2.2
	*S. warneri*	16	2.2
	*S. xylosus*	1	2.2
*Enterococcus* spp.			4.4
*Escherichia coli*			2.2
*Klebsiella pneumonia*			2.2
*Stenotrophomonas maltophilia*			2.2
Others (molds, yeasts, and not other bacteria)			40

**Table 2 microorganisms-11-00709-t002:** Virulence factors in *Staphylococcus* isolates: proteolytic, hemolytic activity, and biofilm formation.

Virulence Factor	Isolates (%)	Prevalence Rates (%)
Protease Production		
Negative (−)	1–3,8,11,12,17,22	36.4
Weak positive (+/−)	6,9,13–15	22.7
Positive (+)	4,5,7,10,18,19,21	31.8
Strongly positive (++)	16,20	9.1
Hemolytic Activity		
-	1,3,7,13–16,20,21	40.9
α-hemolysis	4–6,18,22	36.4
β-hemolysis	2,8–12,17,19	22.7
Biofilm Formation (adherent/planktonic cells)		
<0.75	3,8–10,12,15,17,19	36.4
≥0.75–1	2,7,13,16,20–22	31.8
≥1	1,4–6,11,14,18	31.8

## Data Availability

The data presented in this study are available on request from the corresponding author.

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
