# Peer review of "Short Peripheral Venous Catheters Contamination and the Dangers of Bloodstream Infection in Portugal: An Analytic Study"

_microorganisms, 2023, doi:10.3390/microorganisms11030709_

Round 1

Reviewer 1 Report (Previous Reviewer 2)

The authors have improved the manuscript compared to previous version.

Now the aim of the study is well focused.

The reviewer accepts the current version without modifications.

Author Response

Dear Reviewer 1,

We appreciate the efforts made during the initial revision of the manuscript and feedback provided.

Reviewer 2 Report (New Reviewer)

The manuscript by Nádia Osório and colleagues describes the identification and analysis of microbial species contaminating vascular access devices and causing PIVCs-related infections in Portugal.  This timely manuscript is clearly written, the experiments are adequately controlled, and the subject described in the manuscript fits well within the scope of the journal.

I have only minor suggestions on improvement of the manuscript:

Line 260  -  “was the most common (27.7%)” – in the Table 1, however, the prevalence % for S. epidermidis is 26.7

Line 291 – “membrane” -  needs to be changed to matrix

Author Response

Dear Reviewer 2,

We appreciate the efforts made during the revision of the manuscript and provided feedback. All requests were integrated in the final version (attached).

Best regards

This manuscript is a resubmission of an earlier submission. The following is a list of the peer review reports and author responses from that submission.

Round 1

Reviewer 1 Report

The research is described clearly and completely. The experimentation starts from a well-posed hypothesis and from an excellent experimental design. But the results are the results are poorly presented. Tables or figures are missing. Instead, there are photos that in my opinion could be omitted. 

 I note only small inconsistencies in the text, which I bring to the attention of the authors.

line 20, 27....: please use italics for the name of the genes throughout the text

line 196-238: it is unclear how many Staphylococcus strains was in fact investigated, please describe this clearly by giving the number

line 255: please use italics for the name of bacteria genus

line 298-300: please describe what can be the other mechanics for methicillin resistance

Reviewer 2 Report

In this work the authors reported the microbial contamination of PIVCs, identifying the most prevalent microorganisms, their virulence factors and antibiotic resistance.

The authors  affirm that there are few studies evaluating the contamination of PIVCs, and the importance of prevention.  Thus, in the conclusions they report that the study is innovative at a preventive level, and may play an important role in hospital-acquired infections.

In the opinion of the reviewer this study lacks innovation, since it is well known the risk associated to vascular access devices as for peripheral intravenous catheter (PIVC).

Additionally, as also reported in the references list of the manuscript, there are many studies in the literature describing the mechanism of action of microrganisms and their virulence factors able to favour the attachment to medical devices. Thus, the results obtained are redundant and lack of novelty.

Just as for example, you can find the link to a guideline reported on web illustrating the common procedure for the prevention of peripheral intravenous catheter infections.This guideline has been developed as part of the I-Care intervention bundle for the management of intravascular devices (IVDs). This guideline provides recommendations regarding best practice for the use and management of invasive devices based on current evidence for the prevention and control of healthcare associated infection (HAI)”. Australian Commission on Safety and Quality in Health Care, National Safety and Quality Health Service Standards (September 2012). Sydney. ACSQHC, 2012. Standard 3: Preventing and Controlling Healthcare Associated Infections”.

Guideline: Peripheral intravenous catheter (PIVC)

https://www.health.qld.gov.au › assets › pdf_file